# Fullerenols Prevent Neuron Death and Reduce Oxidative Stress in *Drosophila* Huntington’s Disease Model

**DOI:** 10.3390/cells12010170

**Published:** 2022-12-31

**Authors:** Olga I. Bolshakova, Alina A. Borisenkova, Ilya M. Golomidov, Artem E. Komissarov, Alexandra D. Slobodina, Elena V. Ryabova, Irina S. Ryabokon, Evgenia M. Latypova, Elizaveta E. Slepneva, Svetlana V. Sarantseva

**Affiliations:** Petersburg Nuclear Physics Institute Named by B.P. Konstantinov of National Research Centre “Kurchatov Institute”, 188300 Gatchina, Russia

**Keywords:** fullerenols, *Drosophila melanogaster*, Huntington’s disease, oxidative stress, neurodegeneration

## Abstract

Huntington’s disease (HD) is one of the human neurodegenerative diseases for which there is no effective treatment. Therefore, there is a strong demand for a novel neuroprotective agent that can alleviate its course. Fullerene derivatives are considered to be such agents; however, they need to be comprehensively investigated in model organisms. In this work, neuroprotective activity of C_60_(OH)_30_ and C_120_O(OH)_44_ fullerenols was analyzed for the first time in a *Drosophila* transgenic model of HD. Lifespan, behavior, oxidative stress level and age-related neurodegeneration were assessed in flies with the pathogenic Huntingtin protein expression in nerve cells. Feed supplementation with hydroxylated C_60_ fullerene and C_120_O dimer oxide molecules was shown to diminish the oxidative stress level and neurodegenerative processes in the flies’ brains. Thus, fullerenes displayed neuroprotective activity in this model.

## 1. Introduction

Huntington’s disease (HD) is an autosomal dominant neurodegenerative disorder caused by the mutation in the *huntingtin* (*Htt*) gene which generates a mutant protein (HTT) with an expanded polyglutamine (polyQ). Oligomers accumulation or mutant HTT aggregates leads to the death of striatal and cerebral cortex neuron populations. PolyQ expansion is believed to cause abnormal β-sheets formation in protein structure with increased stability due to hydrogen bonding between the main and side chain amides [1,2]. During mutant protein folding, mechanisms targeting misfolded proteins for their eventual degradation by molecular chaperones, as well as ubiquitination and autophagy, cease to function [3]. This negatively influences transcription, energy metabolism and mitochondria, increases reactive oxygen species (ROS) and inflammation processes, and eventually leads to cell damage [4,5,6,7]. Moreover, the mutant HTT is thought to affect some cytoskeleton components, resulting in axonal transport disruption and cell death [5].

Oxidative stress (OS) is one important HD pathogenesis environment. Its negative role has been described both in studies of HD patients and in experimental models [6,8,9]. Therefore, the study of the role of OS and the search for new effective antioxidants is key to developing a complex treatment of HD and other neurodegenerative diseases (ND) [10]. 

Fullerenes and their derivatives are powerful antioxidants due to their electron deficiency property accounted for by interaction with free radicals; therefore, they are called “Radical sponges” [11]. A number of studies have experimentally confirmed their antioxidant activity [12,13]. Moreover, fullerenes have been shown to prevent amyloid proteins aggregation [14,15,16]. The major or dominant factors of HD pathogenesis have not been identified so far. Perhaps they are all complementary and interdependent [4,5]. The unique properties of fullerenes and their derivatives open the possibility of not only correcting some aspects of pathology development, but also studying the role of individual components in this process on model organisms. At the same time, fullerenes hydroxylated derivatives show greater compatibility with biological systems than do native fullerenes; in addition, they have low toxicity [17] and leave the body through the urinary tract [18,19,20].

The aim of this work was to analyze the C_60_(OH)_30_ and C_120_O(OH)_44_ fullerenols in terms of its neuroprotective activity on a *Drosophila* HD model. *Drosophila melanogaster* is a good model to study diseases with relatively simple genetic backgrounds such as HD [21,22]. It can be used to study the effects of an increase in the CAG repeats number, as well as the influence of genetic modifiers and environmental factors on the disease pathogenesis [23,24,25]. Using *Drosophila* as a model organism makes it possible to study and modulate tissue-specific genes expression, test existing and novel hypotheses in pathology development and take into account a large number of animals [26,27,28]. The latter is very important, as it allows for drugs primary screening, which significantly speeds up, and reduces the cost of, the discovery and development of new drugs. 

## 2. Materials and Methods

### 2.1. Fullerenols

Fullerenols C_60_(OH)_30_ and C_120_O(OH)_44_ were obtained at the Petersburg Nuclear Physics Institute, named by the B.P. Konstantinov of National Research Center as the “Kurchatov Institute” [17,29]. The structure of fullerenols is qualitatively identified by Fourier infrared spectroscopy in the range 4000–400 cm^–1^ using a spectrometer IRTracer-100 (Shimadzu) and MALDI-TOF/TOF mass spectrometry (Axima Resonance Mass Spectrometer—Shimadzu, Kyoto, Japan). Dynamic light scattering and zeta potential experiments were performed on a Zetasizer nano (Malvern Panalytical, Malvern, UK) size analyzer. The measurements were carried out three times. The data were collected at 25 °C by monitoring the scattered light intensity at a 90° detection angle. Initial concentrations of fullerenols were 20 mg/mL for C_60_(OH)_30_ and 1.05 mg/mL for C_120_O(OH)_44_. The optimal experimental concentrations were identified based on toxicity and antioxidant activity (AA) data in vitro.

### 2.2. In Vitro Toxicity Analysis of Fullerenols

For this study, we used ECV cells (human umbilical vein endothelial cells), A549 cells (human lung adenocarcinoma cell) and Hela cells (human cervical carcinoma cell) (National Research Center “Kurchatov Institute”—PNPI collection). Cells were cultured in DMEM with L-Glutamine (Capricorn Scientific, Ebsdorfergrund, Germany), antibiotics (penicillin and streptomycin, Biolot, Saint-Petersburg, Russia) and 10% bovine serum (Biolot, Saint-Petersburg, Russia). Cells were incubated at 37 °C in 5% CO_2_. Toxicity was determined using a standard MTT test (13). The culture medium was replaced with one containing C_60_(OH)_30_ (0.2 and 0.02 mg/mL) or C_120_O(OH)_44_ (0.01 and 0.001 mg/mL) a day after cell seeding. After 24 h, an MTT test was performed. The absorbance was determined at a wavelength of 540 nm using a Multiskan FC microplate photometer (Thermo Scientific, Waltham, MA, USA).

### 2.3. In Vitro Antioxidant Activity Assay

Antioxidant activity (AA) of fullerenols in vitro was measured by their ability to inhibit adrenaline autoxidation, according to [30,31] with modifications. The reaction mixture contained 100 μL 0.1% adrenaline hydrochloride solution and 2 mL 0.2M sodium carbonate buffer (pH 10.7). The mixture was mixed quickly and placed in a spectrophotometer. The absorbance was determined at 347 nm every 30 s for 15 min. Then, 2 mL 0.2 M sodium carbonate buffer, 50 μL of the investigated fullerenol solution and 100 μL 0.1% adrenaline hydrochloride solution were mixed. The absorbance was determined as described above. A similar mixture, but not containing adrenaline, was placed in the comparison cuvette. AA was expressed as the percentage of inhibition of adrenaline autoxidation and was calculated by the formula:(1)$$\mathrm{AA}=\frac{\left({\mathrm{Abs}}_{1}-{\mathrm{Abs}}_{2}\right)\times 100}{{\mathrm{Abs}}_{1}}\;\%$$
where Abs_1_ is the absorbance of adrenaline, and Abs_2_ is the absorbance of adrenaline with fullerenol. A value greater than 10% indicates significant AA.

The optical absorption spectra of adrenaline autoxidation products were recorded on an SF-2000 spectrophotometer (LLC OKB Spectr, Saint-Petersburg, Russia). The root-mean-square error of the arithmetic mean optical density for three experiments did not exceed 10%.

### 2.4. In Vivo ROS Measurement

Oxidative stress was measured using the H2DCF-DA probe. A total of 20 flies’ heads were used for the lysate preparation. The heads were homogenized in a mixture containing 100 μL of 10 mM Tris and 3 μL of protease inhibitor Cocktail tablets (Roche, Penzberg, Germany) (pH = 7.4). The homogenate was centrifuged for 10 min at 10,000 rpm. A total of 5 μL of lysate and 60 μL of 5 μM H2DCF-DA (Invintrogen, Waltham, MA, USA) were dropped into a well of a 96-well plate and were incubated for 60 min at 37 °C. The signal was measured on a flatbed analyser with λwt = 490 nm, λem = 530 nm, and h = 4.7 mm. The signal was normalized to protein concentration measured by the Bradford method.

### 2.5. Drosophila Lines

All strains were obtained from the Drosophila Bloomington Stock Center (Bloomington, IN, USA). Flies were maintained on standard yeast medium in 12-h/12-h light/dark photoperiod at 25 °C. Transgene expression was carried out in the GAL4/ UAS binary system [32]. Transgenic *Drosophila* strains related to the current study are listed here: *UAS-HTT.16Q.FL* (contains *Htt* human gene, encoding HTT human protein with a normal PolyQ repeats—16) (RRID:BDSC_33810); *UAS-HTT.128Q.FL* (contains *Htt* human gene, encoding HTT human protein with an expanded PolyQ repeats—128) (RRID:BDSC_33808); *elav-GAL4*, a strong neuronal cell-specific driver (RRID: BDSC_458); *Cha-GAL4*, *UAS-GFP.S65T*, GFP expression in all cholinergic neurons (RRID:BDSC_6793).

### 2.6. Drosophila Lifespan Experiment and Climbing Assay

Flies were placed in test tubes with agar (30 flies, male and female, per tube) at 25 °C. The flies were fed with yeast inactivated at 65 °C with the addition of 0.2 mg/mL of fullerenols C_60_(OH)_30_, or 0.01 mg/mL C_120_O(OH)_44_. First, yeast was thoroughly mixed with nanoparticles using vortex, and then 80 μL of feed was dripped onto the agar surface, which is not a nutrient medium for *Drosophila*. Every two days, living flies were transferred to fresh food. Simultaneously, the number of dead specimens was counted. Males and females were counted separately. The observation was carried out until the last dead fly was detected, and the maximum lifespan was determined. There were at least 300 flies in each experiment. Each experiment was run using triplicates.

The climbing analysis was carried out on the flies’ offspring, obtained from crossing between *UAS-HTT.16Q.FL* and *elav-GAL4* (further, *HTT.16Q*), or *UAS-HTT.128Q.FL* and *elav-GAL4* (further, *HTT.128Q*) on days 5, 15 and 25 of life, according to the method [17].

### 2.7. Analysis of Neurodegeneration

Brain preparations for confocal microscopy were prepared according to the method [33]. A series of images were obtained using a Leica TCS SP5 microscope (Leica, Wetzlar, Germany) with a built-in 35-mW argon laser at a wavelength of 488 nm. The shooting parameters were an optical thickness of 2 µm and image resolution of 1024 × 1024 and 40× (oil) objective. The cells number of interneurons (IN), belonging to the cholinergic neurons (CN), was counted on 3D projections in Leica Application Suite X and ImageJ.

The total neurodegeneration level was assessed in paraffin brain sections. Fly heads were fixed in 3.7% paraformaldehyde (Sigma-Aldrich, Saint Louis, MO, USA) for 24 h, embedded in paraffin, and 6 μm brain slices were stained with hematoxylin/eosin (Bio Optica, Milano, Italy). The paraffin slices were examined using a Leica DM 2500 microscope. To estimate the extend of degeneration, we calculated the ratio between areas free of cells and the total area of the brain.

### 2.8. Immunoblotting

Western blot analysis was carried out according to the method, which is described in detail in the article with the modifications [33]. A total of 100 heads were taken for each experiment. Blots probed using the primary Anti-Huntingtin Protein antibody, a.a. 181–810, clone 1HU-4C8 (Millipore Cat # MAB2166) or mouse monoclonal anti–b–tubulin (Santa Cruz Biotechnology Cat # sc-365791, RRID: AB_10841919) antibodies were diluted in BlockPRO buffer. Samples with antibodies were incubated at 4 °C overnight.

Bands were quantified using ImageJ software (NIH, Bethesda, MD, USA). The density of the background staining was subtracted from all signals, and the densities of the 348 kDa Huntingtin bands were normalized to the densities of the tubulin bands (55 kDa).

### 2.9. Statistical Analysis

Statistical analysis was performed using KyPlot 6.0 software. The Dunnetts-test was used for a comparison of two samples (toxicity, in vivo OS analysis, climbing assay and total neurodegeneration) and the Tukey-Kramer was used for multiple comparisons (to analyze the IN number). Differences at *p* < 0.05 were considered statistically significant.

## 3. Results

### 3.1. Toxicity Analysis and Antioxidant Activity of Fullerenols In Vitro

The fullerenes synthesis and characterization used in the work have previously been described in detail [17]. As shown in Appendix A, the four intramolecular vibrational characteristic modes T_1u_ of pristine C60 (A) at 526 cm^−1^, 576 cm^−1^, 1184 cm^−1^ and 1428 cm^−1^ disappear from the FTIR spectra of dimer oxide C_120_O (D) and fullerenols (B, C). The dimer oxide C_120_O contains characteristic bands at ~800 cm^−1^, 1028 cm^−1^, 1080 cm^−1^, 1261 cm^−1^ [34]. The fullerenols spectra (B, C) has broad hydroxyl absorption around 3390 cm^−1^, a C—O stretching absorption at 1080 cm^−1^ and C = C absorption at 1618 cm^−1^. The peak centered at 1370 cm^−1^ can be assigned to bending vibrations of hydroxyl groups. IR spectra of fullerenols also showed the appearance of a C = O stretching peak at 1720 cm^−1^.

The mass spectrum of C_60_(OH)_30_ (Appendix A) showed peaks with a high intensity at m/z = 720, which demonstrates that the fullerene cage has not been damaged during fullerenol synthesis. Detection of a molecular ion at m/z 1230 was indicative for the composition of fullerenol as containing 30 hydroxyl groups per C_60_ cage. Fullerenols under the conditions of MALDI-TOF analysis lose their hydroxyl groups, and the use of various matrices (dihydroxybenzoic acid, dithranol) did not change the results. A similar situation is observed in the mass spectrum of C_120_O(OH)_44_ fullerenol (Appendix A). In addition to the detachment of hydroxyl groups, defragmentation of the initial oxide molecule of the C_120_O dimer also occurs, with successive detachment of two carbon atoms from the cages, as evidenced by the appearance of lines in the spectra that differ by 24 Da.

Previously, we found that fullerenols C_60_(OH)_30_ and C_120_O(OH)_44_ in the concentration range of 0.01–1.0 mg/mL (C_60_(OH)_30_) and 0.005–0.1 mg/mL (C_120_O(OH)_44_) in an aqueous solution form polydisperse aggregates without a clear dependence on the concentration of particle sizes and zeta potential [17]. Appendix A shows the zeta potential and size distribution for fullerenols C_60_(OH)_30_ and C_120_O(OH)_44_ at concentrations of 2 mg/mL and 0.1 mg/mL, respectively.

As can be seen from Appendix A, there are aggregates of two sizes in the fullerenol solution—144 ± 44 nm and 814 ± 143 nm for C_60_(OH)_30_ and 14 ± 4 nm and 516 ± 45 nm in C_120_O(OH)_44_ solution. Zeta potential was negative with the mean value of −15.2 ± 6.3 mV for C_60_(OH)_30_ and −28.8 ± 6.3 mV for C_120_O(OH)_44_.

According to modern concepts, all newly synthesized carbon nanoparticles derivatives should be tested for toxicity. Fullerenols synthesized for this study were analyzed using the MTT test. C_120_O(OH)_n_ was additionally purified from traces of unreacted hydrogen peroxide during the hydroxylation reaction, since in previous work C_120_O(OH)_n_ showed greater toxicity than C_60_(OH)_30_ [17]. Also, the OH groups number in C_120_O(OH)_n_ was determined to be 44 ± 2 using the thermal estimation method of the substituents number [35]. It has been shown that C_60_(OH)_30_ at the studied concentrations is safe for ECV, A549 and Hela cells. C_120_O(OH)_44_ was non-toxic to ECV and A549 cells and Hela cell viability was reduced by only 10–15% (Figure 1A). Changes in cell morphology were not observed when exposed to nanoparticles.

It is known that the attachment to the fullerene cage of even one addend can lead to a significant change in its reactionary ability, including in relation to free radicals. Appendix A presents the absorption spectra of autoxidation products of pure adrenaline and adrenaline in the presence of C_60_(OH)_30_ in a 0.2 mg/mL concentration and C_120_O(OH)_44_ in a 0.01 mg/mL concentration 5 min after the reaction started. As can be seen from Appendix A, the C_60_(OH)_30_ and C_120_O(OH)_44_ addition does not lead to the appearance of a new absorption peak, while the absorbance value of the intermediate product of adrenaline autoxidation is reduced at 347 nm. A similar situation is observed with the addition of C_120_O(OH)_44_ to adrenaline. It is assumed [36] that the fullerenols do not directly interact with adrenaline and its oxidation products, and the inhibiting of the adrenaline autoxidation reaction is achieved by the competing reaction of the fullerenols’ interaction with the superoxide anion radicals (O^2−^˙) that occur during intramolecular restructuring of the adrenaline molecule, as Appendix A shows. Superoxide radicals can join *sp^2^*-hybridized carbon atoms with the formation of fullerenol anion-radical. The water present in the system then gives a proton to a superoxide radical with the formation of the OH^−^ and HOO˙ groups on the carbon cage. Then these two stages are repeated and two HOO˙ groups on the carbon cage are isomerized with the formation of oxygen and hydrogen peroxide. The schemes of these processes are shown in Appendix A).

Figure 1B,C shows the dependence of the absorbance of the intermediate product of adrenaline oxidation on the reaction time of adrenaline autooxidation in the absence and in the presence of fullerenols. The results presented in Table 1 reveal that the AA of the fullerenols depends on the concentration being non-linear. The AA change during the adrenaline autoxidation is associated with the transition of the intermediate products of the reaction with an absorption maximum at 347 nm to adrenochrome with an absorption maximum at 480 nm. Therefore, it is more reliable to evaluate AA at a time of 10 to 15 min from the start of the adrenaline autoxidation reaction, when the rate of intermediate products formation is equal to the rate of their transition to adrenochrome.

### 3.2. Lifespan and Behavior Analysis of Drosophila in a Huntington’s Disease Model

Characteristic features of HD are reduced lifespan and movement disorders. To model HD, we activated HTT neuronal-specific expression with normal (*HTT.16Q*) and expanded (*HTT.128Q*) polyQ repeats in flies using GAL4/UAS binary system. The lifespan of *HTT.128Q*-expressing flies was reduced by more than half compared to those expressing *HTT.16Q* (Figure 2A). Sex was taken into account when counting the dead flies. The lifespans of males and females with *HTT.16Q* was equal, while in the *HTT.128Q* flies, the lifespan of males was shorter than that of females. Locomotor activity was monitored by a climbing assay (negative geotaxis analysis). The ability to climb decreased with age in the *HTT.128Q* flies at a significant rate compared to the *HTT.16Q* flies (Figure 2B), and was minimal on the 25-day-old flies. We did not observe a significant difference between the males’ and females’ locomotor activity, and thus combined them in the analysis in a 1:1 ratio.

We tested the fullerenols effects on lifespan and locomotion activity on this model, since the *HTT.128Q* neuronal-specific expression in flies led to a violation of the main vital signs. The lifespan and locomotion activity of males and females with *HTT.128Q* did not decrease when C_60_(OH)_30_ at a dose of 2 mg/mL or 0.2 mg/mL and C_120_O(OH)_44_ at a dose of 0.1 mg/mL or 0.01 mg/mL were added to the feed (Figure 3A–C). Thus, fullerenols were proven to be safe for flies.

### 3.3. Analyses of Oxidative Stress

Next, we assessed the level of ROS in the *HTT.128Q* fly brain using 2’,7’-dichlorodihydrofluorescein diacetate. As shown in Figure 3D, we found that ROS levels were higher in the brains of the 15-day-old and 25-day-old *HTT.128Q* flies compared to the control flies. Fullerenols are active scavengers of ROS, unlike some other antioxidants imitating the activity of antioxidant defense enzymes. Therefore, we analyzed the effect of fullerenols on the level of ROS in the brain of *HTT.128Q* flies. The content of *Drosophila* on the medium, including fullerenols (C_60_(OH)_30_ at a dose of 0.2 mg/mL or C_120_O(OH)_44_ at a dose of 0.01 mg/mL), led to a significant decrease in the ROS level in 15-day-old flies.

### 3.4. Analysis of Neurodegeneration

It is well known that oxidative stress is a hallmark of neurodegeneration, although the relationship between cell death and the accumulation of reactive oxygen species remains unclear [37]. We performed histological analyses of brain sections stained with hematoxylin and eosin on days 15 and 25–30 of the fly’s adult life (Appendix A). We observed an increase in vacuolization and overall neurodegeneration in the brain of *HTT.128Q* flies with age (Figure 4A). These parameters were corrected by adding C_60_(OH)_30_ and C_120_O(OH)_44_ fullerenols to food. The vacuoles area on the brain sections was less in the 15-day-old *HTT.128Q* flies, fed with fullerenols, unlike the control group. However, on the 25th day, a statistically significant slowdown in neurodegeneration was observed only when fullerenol C_120_O(OH)_44_ was added to the flies’ food (Figure 4A).

It has previously been shown that during the development of HD, mainly the GABAergic and cholinergic systems are depleted [5]. At the same time, cholinergic neurons are the most numerous and play an important role in *Drosophila* [38]. Therefore, we expressed *HTT.128Q* in acetylcholinergic neurons and determined their number on days 5, 15 and 25 by confocal microscopy. Intermediate cholinergic neurons (INs) were selected for analysis, which are well distinguishable in the fly brain (Figure 4C). It was shown that the INs number in *HTT.128Q* flies decreases with age (Figure 4B). Adding fullerenols to food for 15 days prevented neuron death and increased numbers in the brain. However, at 25 days old, the effectiveness of fullerenols dropped (Figure 4B,C) and there was no statistically significant difference between the control group and the *HTT.128Q* flies.

### 3.5. Analyses of HTT Level

Human HTT was detected in the fly brain using the HTT-specific antibody MAB2166. We detected a band corresponding to the full-length protein (∼350 kDa). We did not detect insoluble HTT, which is consistent with data obtained by other researchers [39]. The soluble HTT level in the fly brain did not differ between those who received normal food and food with fullerenols (C_60_(OH)_30_ at a dose of 0.2 mg/mL or C_120_O(OH)_44_ at a dose of 0.01 mg/mL) for 30 days (Figure 5A,B).

## 4. Discussion

The various HD models differing in the length of polyglutamine repeats have been created in *Drosophila* to study pathogenesis and search for potential therapeutic agents. The flies expressing elongated polyQ repeats imitate HD in terms of reduced lifespan, reduced motor activity, photoreceptor degeneration, etc. [40]. It has been shown that the pathological effects of HTT are not necessarily associated with its aggregation. Thus, Akbergenova Y et al. [28] revealed that pathogenic HTT can be present in a significant amount in *Drosophila* synapses in a non-aggregated form, disrupting synaptic growth. Its accumulation in neuromuscular junctions is stochastic. At the same time, increased mortality of the third instar larvae has been observed. Romero E et al. also used a model with no accumulation of mutant HTT aggregates [39]. They have demonstrated that the expression of *HTT.16Q*, as well as *HTT.128Q*, resulted in a full-length ~350 kDa human protein accumulation in the flies’ heads. The protein has been located in the cytoplasm and did not form aggregates in axons. At the same time, *HTT.128Q* expression was associated with the progressive degeneration of photoreceptor cells and degenerative phenotypes in the peripheral nervous system, as well as with those correlated with them: early death, behavioral disorders and impaired electrophysiological parameters. According to the authors, neurodegeneration in this case is based on Ca^2+^-dependent increased neurotransmission in the early stages of pathogenesis, caused by an increased CAG triplets number in the protein.

Using the model of Romero et al. [39], we obtained similar results: a decrease in life expectancy, impaired behavior and neurodegeneration against the background of the formation of only soluble HTT forms. We also observed ROS level increase in the brains of *HTT.128Q* flies with age. According to modern concepts, OS is a significant factor leading to the emergence of many NDs and one of the central nonspecific pathogenesis components [41]. There is evidence that mitochondrial disfunctions and energy metabolism defects in the CNS are important in triggering HD [8]. The mutant HTT is known to change the PGC-1α factor functional activity after binding. The PGC-1α factor regulates the genes expression of the main antioxidant enzymes SOD1, SOD2 and Gpx-1 [42]. Impairment of antioxidant defense system leads to ROS excessive formation, causing lipid peroxidation, cell structures, proteins, DNA, RNA damage and mitochondrial dysfunction [41]. Altered biomolecules themselves become OS sources and a vicious circle arises [43]. Severe and prolonged OS can trigger apoptosis and necrosis, and, accordingly, lead to selective damage to neurons [8].

The neuroprotective and antioxidant activity of carbon nanoparticles, in particular fullerenes and their derivatives, has been shown in studies in vitro [44,45,46], as well as in vivo models of Alzheimer’s, Parkinson’s, etc. [47]. We have previously shown the neuroprotective effect of fullerenols on the models of Parkinson’s disease and Alzheimer’s disease in *Drosophila*. In the first case, C_60_(OH)_30_ and C_70_(OH)_30_ reduced OS and the level of the alpha synuclein insoluble form in the flies’ brains, and prevented dopaminergic neurons death [33]. In the second case, C_60_(OH)_30_ prevented the aggregation of amyloid peptide beta, and reduced the ROS number and the overall neurodegeneration level in the brain [48]. The perspectives of fullerene applications for HD treatment have been broadly discussed in the literature, but we are aware of only one experimental work devoted to this problem [49].

In this work, we have shown that fullerenols C_60_(OH)_30_ at a dose of 0.2 mg/mL or C_120_O(OH)_44_ at a dose of 0.01 mg/mL led to a significant decrease in the level of ROS in the brain of 15- and 25-day-old *HTT.128Q* flies. However, the decrease in the level of ROS maximally correlated with the slowing down of neurodegeneration only on the 15th day, and the effectiveness of fullerenols in preventing neuronal death decreased on the 25th day. Our results suggest that OS plays a considerable, but not dominant, role in neuronal death. The vital function for neurons is mediated by HTT: it promotes the movement of various vesicles along axons, acting as a connecting element (framework) between motor proteins, “cargo” and microtubules. Mutations in *Htt* disrupt the organelles transport, including mitochondria, into axons. At the same time, abnormal forms of the protein have been shown to interact with the Drp-1 protein regulating mitochondria division. As a result, the accumulation of fragmented organelles takes place, followed by apoptosis activation [50]. Note that we did not observe a large difference in the effects of C_60_(OH)_30_ and C_120_O(OH)_44_, which may be due to the comparable sizes of their particles and zeta potential [17].

We were unable to affect the lifespan and locomotor activity of flies with the help of fullerenols. But it should be noted that the integral indicator of vital activity—lifespan of *D. melanogaster*—is highly conservative and is not always amenable to correction by drugs. The test for locomotor activity also did not show a correlation with a decrease in neuron death. We assume that in our model, as in cases of ND in humans, the ROS level reaches critical values with age, and the antioxidants used are no longer able to prevent neuronal death or locomotor impairments or increase life expectancy. According to Forman and Zhang [51], antioxidant enzymes play an even larger role in antioxidant protection than ROS scavengers. That may explain why exogenous antioxidants demonstrate limited effectiveness.

Thus, using the *Drosophila* HD model, we were able to demonstrate for the first time that C_60_(OH)_30_ and C_120_O(OH)_44_ fullerenols have antioxidant and neuroprotective activity. Although the involvement of oxidative stress in the pathogenesis of HD is now well established [52], it is still not clear whether oxidative stress is a cause, a consequence or simply a by-product of the neurodegenerative process that occurs in the brains of HD patients. In addition, Huntington’s disease is currently considered a multifactorial disease, which implies the creation of a complex therapy that affects various cellular processes, part of which could be water-soluble fullerenols. However, further investigations are required to elucidate the detailed mechanism of fullerenols activity.

## Figures and Tables

**Figure 1 cells-12-00170-f001:**
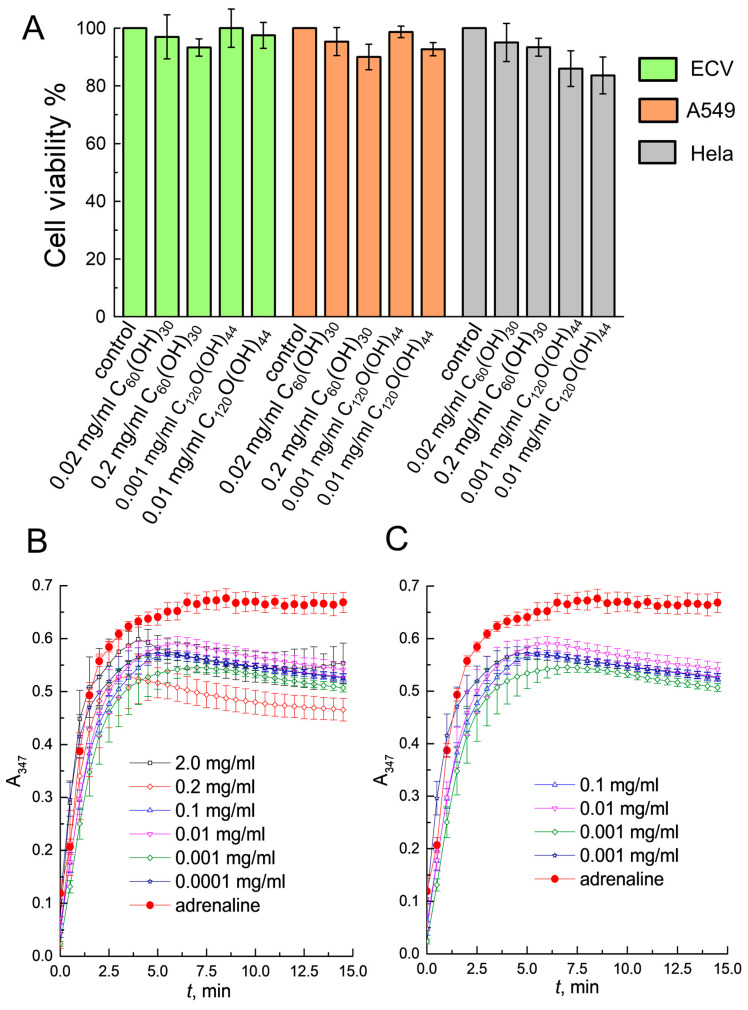
In vitro cytotoxicity and antioxidant analysis of fullerenols. (**A**) MTT assay. Fullerenols were added to the culture 24 h after seeding and were present in the culture medium for 24 h. Control cells without fullerenols. The difference between the control sample and experimental sample is unreliable (*p* ≥ 0.05). *n* ≥ 6 experiments. (**B**,**C**) Absorbance value dependence vs. reaction time adrenaline autoxidation; (**B**) in absence and presence of C_60_(OH)_30_, mg/mL; (**C**) in absence and presence of C_120_O(OH)_44_, mg/mL. The errors on the curves (**B,C**) are the mean ± SEM (*n* = 5).

**Figure 2 cells-12-00170-f002:**
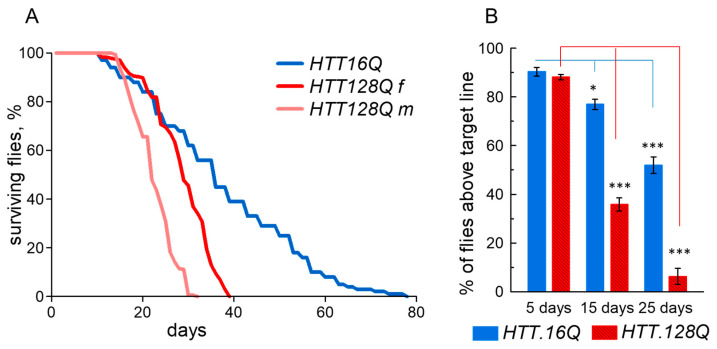
Comparison of lifespan (**A**) and locomotion activity for 5-, 15- and 25-day-old flies (**B**) with neuronal-specific expression of *HTT.16Q* and *HTT.128Q*. The Dunnetts-test, mean ± SEM. 20–60 flies per point, *n* = 3 separate experiments, *** *p* < 0.001; * *p* < 0.05.

**Figure 3 cells-12-00170-f003:**
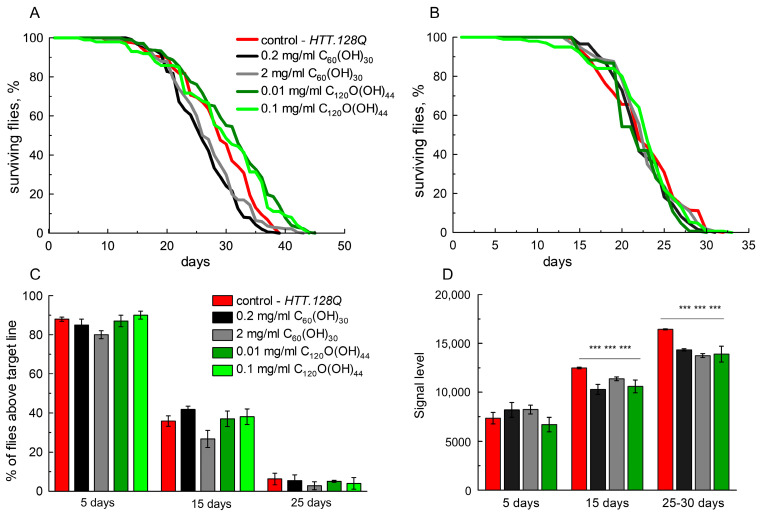
The influence of C_60_(OH)_30_ and C_120_ O(OH)_44_ fullerenols of different concentrations on vital signs of flies expressing *HTT.128Q*. (**A**) Lifespan of male; (**B**) lifespan of females; (**C**) locomotor activity analysis at different ages (mixed sex flies: male and female). The Dunnetts-test, 20–100 flies per point, *n* = 3 separate experiments; (**D**) fluorescent signal level corresponding to the total reactive oxygen species (ROS) concentration in brain samples of flies at different ages. Tukey–Kramer test, mean ± SEM is shown. 50 heads per point, *n*= 3 separate experiments, *** *p* < 0.001.

**Figure 4 cells-12-00170-f004:**
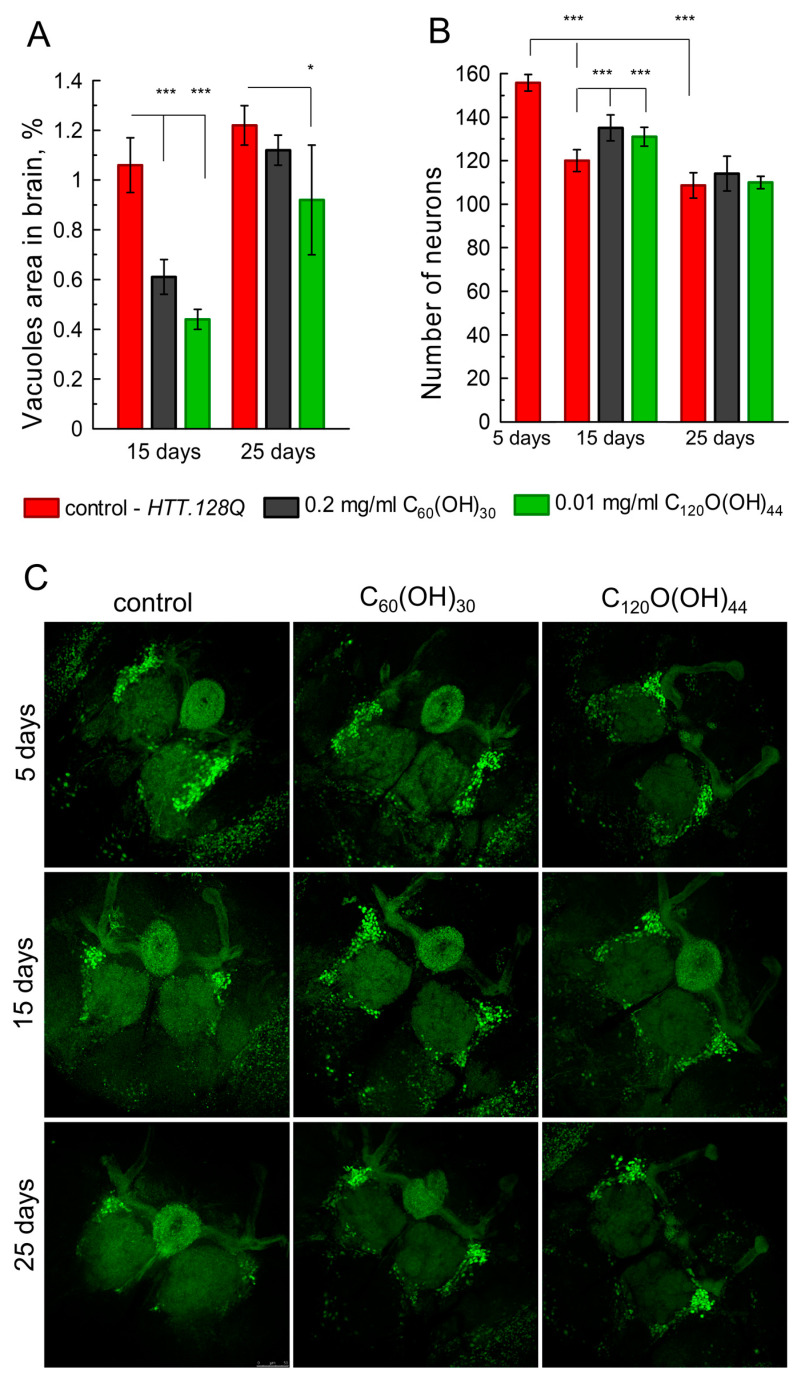
Neurodegeneration analysis. (**A**) The total hole area in fly brains with *HTT.128Q* expression in all neurons on 15 and 25 days old. The Dunnetts-test, mean ± SEM. N = 6 per point; (**B**) the INs number with *HTT.128Q* expression in cholinergic neurons at different ages. Tukey–Kramer test, mean ± SEM is shown. N = 12 per point *** *p* < 0.001, * *p* < 0.05; (**C**) female IN confocal images at different ages. Green: GFP in IN (*Cha-GAL4, UAS-GFP.S65T*). Scale bar: 50 µm.

**Figure 5 cells-12-00170-f005:**
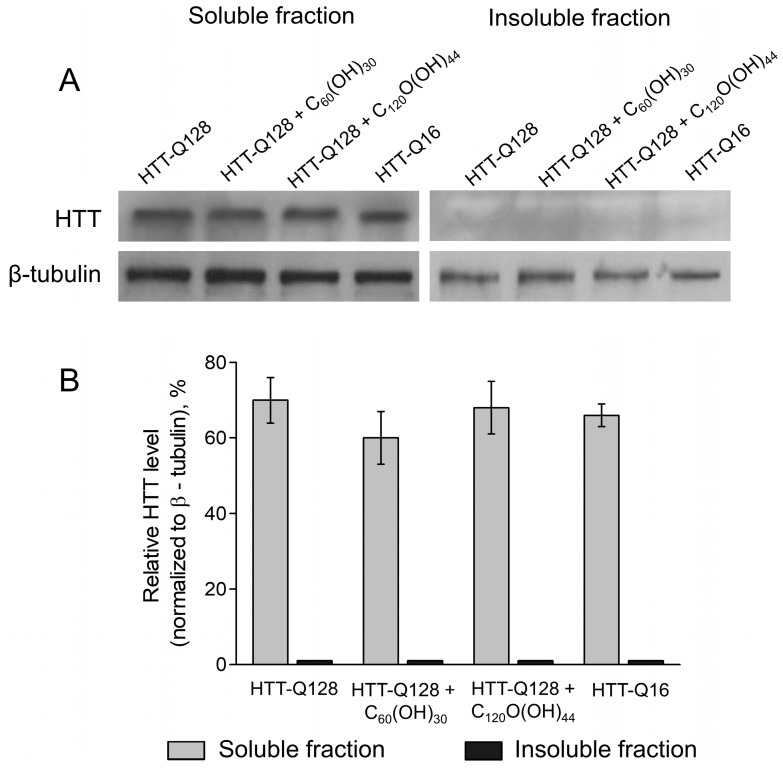
Analyses of HTT level. (**A**) Representative Western Blot showing the levels of soluble and insoluble HTT in whole brain lysates of *Drosophila melanogaster*; (**B**) the bar graph showing the influence of fullerenols on the level of soluble HTT. Tukey–Kramer test, mean ± SEM is shown. N = 100 for each experiment.

**Table 1 cells-12-00170-t001:** Antioxidant activity of Fullerenols. Dependence of the effect on the reaction time and concentration of fullerenes.

C_120_O(OH)_44_ mg/mL	AA, %			C_60_(OH)_30_, mg/mL	AA, %		
	Reaction Time, Min				Reaction Time, Min		
	5	10	15		5	10	15
				2	9.2 ± 0.6	18.4 ± 0.7	17.2 ± 0.9
				0.2	19.4 ± 1.1	28.4 ± 1.0	30.4 ± 1.7
0.1	11.6 ± 0.4	18.4 ± 0.9	21.4 ± 0.9	0.1	11.6 ± 0.7	18.4 ± 0.8	21.4 ± 1.2
0.01	8.9 ± 0.3	15.6 ± 0.8	19.0 ± 0.8	0.01	8.8 ± 0.5	15.6 ± 0.6	19.0 ± 1.0
0.001	16.6 ± 0.8	20.5 ± 1.0	24.2 ± 1.0	0.001	21.6 ± 0.8	20.5 ± 0.8	24.2 ± 0.8
0.0001	10.7 ± 0.4	18.3 ± 0.9	21.2 ± 0.9	0.0001	28.2 ± 1.0	18.3 ± 0.7	21.2 ± 1.2

Mean ± SEM.

## Data Availability

The data presented in this study are available in the article and Appendix A.

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
