# Peer review of "Fullerenols Prevent Neuron Death and Reduce Oxidative Stress in Drosophila Huntington’s Disease Model"

_cells, 2022, doi:10.3390/cells12010170_

Round 1

Reviewer 1 Report

The paper entitled "Fullerenols prevent neuron death and reduced oxidative stress in Drosophila Huntington's disease model" provides us more insights about how fullerenols act on the Huntington's disease model in Drosophila. By testing the effect of fullerenols on ROS levels, vital signs of flies and neuron degeneration, they showed the possibility of using fullerenols as part of the treatment for Huntington's disease. 

This paper provides clear evidences of how fullerenols can influence ROS levels. However, there are several questions that I would like the author to address. First, why picked such a low concentration for C120O(OH)44 when the higher concentration showed the similar cell viability but higher antioxidant activity?  Since administration of both C120O(OH)44 and C60(OH)30 in low concentration failed to prevent neuron death on day 25, I would recommend to test the effect of both drugs with higher dose. Also, for the HTT experiments (Figure 5), the author claimed that they didn't see any difference in the soluble HTT levels in the flies brain with or without taking fullenrenols after 30 days. But based on the Figure 3A and 3B, most of flies expressing HTT.128Q were died around 30 days especially the females. Please provide more details about how you performed those experiments when most of flies should be died and why picked this time point. 

Besides, some of the labeling and figure legends are confusing. For example, in figure 1, you didn't tell readers what the error bar represented and the numbering systems are confusing.  There are also some typos and mislabeling in the paper. Please correct them. 

Author Response

We appreciate greatly the comments obtained from reviewer, which made our manuscript better. Please find below our responses (red font) to the reviewers’ comments together with the revised versions of the manuscript and Supplementary Materials. The changes in the manuscript are marked with “Track Changes" function in Microsoft Word, so that changes are easily visible. We hope strongly that the modified manuscript fits completely the standard of Cells and is now suitable for publication.

Reviewer 2 Report

cells (ISSN 2073-4409)

Review of an article: 

«Fullerenols prevent neuron death and reduced oxidative stress in Drosophila Huntington’s disease model»

by Olga I. Bolshakova, Alina A. Borisenkova, Ilya M. Golomidov, Artem E. Komissarov, Alexandra D. Slobodina, Elena V. Ryabova, Irina S. Ryabokon, Evgenia M. Latypova, Elizaveta E. Slepneva, Svetlana V Sarantseva

In Cells (ISSN 2073-4409).

Round 1

The article is concerned with an experimental investigation of finding a novel neuroprotective agent. Fullerene derivatives are considered to be such agents; however, they need to be comprehensively investigated in model organisms. In this work, the neuroprotective activity of С60(ОН)30 and С120О(OH)44 fullerenols was analyzed for the first time in Drosophila transgenic model. Feed supplementation with hydroxylated C60 fullerene and C120O dimer oxide molecules diminished the oxidative stress level and neurodegenerative processes in the fly’s brain. 

The article was presented in a well-structured manner, with a good level of organization. Unfortunately, several statements within have weak evidence. Therefore, the referee suggested that the manuscript be improved with a major revision. The following is a list of specific concerns.

1.     As for the Introduction section, the authors should be used more specific and detailed work since 2022. Only 8 of 41 references belong to the last three years.

2.     Line 81, 94, and 95, 192: The optical density was determined at a wavelength of 540 nm using; where D1 is the optical density of adrenaline; D2 is optical density…

The use of the term optical density is discouraged. It should be used A1 or Abs1. (PAC, 1996, 68, 2223. (Glossary of terms used in photochemistry (IUPAC Recommendations 1996)) on page 2257)

Line 187: “the spectra of optical absorption” should be as “absorption spectrum” https://goldbook.iupac.org/terms/view/A00043

3.     The authors should provide the DLS datum to validate the size distributions. As well, zeta potential should be provided to estimate the stability of suspensions.

4.     Please, provide information concerning the structure of C60(OH)30 and dimeric [60]fullerene derivative C120O(OH)44 and their full description and characterization by MALDI-TOF-MS, for instance. For studying such behaviors, it is essential to understand the surface functional groups on particle dispersion.

5.     It is highly recommended to compare your results with Gd@C82Ox(OH)y https://doi.org/10.3390/ijms23095152and Gd@C82, C70, C60 https://www.mdpi.com/1422-0067/22/11/5838, and https://www.mdpi.com/1422-0067/22/11/6130.

6.     The discussion section needs to be included elucidation concerning the structure of C60(OH)30 and C120O(OH)44and their influence on the obtained results.

7.     For the 3.1 section, provide a kinetic scheme of reactions that will make certain clarity in Table 1 results.

Style guide:

·      Line 311. Please, use the superscript character for Ca2+ >> Ca2+.

·      Line 198 sp2-hybridized should be sp2-hybridized.

English spelling should be double-checked.

Author Response

(The authors gave the same response as above.)

Reviewer 3 Report

The authors tested the effect of fulleronols on survival and neurodegeneration of HTT.128Q expressing flies. The data show that 0.2 mg/ml C60 decreases while 0.01 mg/ml C120 increases the survival of elav>HTT.128Q flies on male, but no effect on female. The chemicals do not have an beneficial effect on movements. The chemicals reduced ROS. The vacuoles area are reduced by fulleronol feeding but with no brain staining shown. Lots of typo-errors suggest that the authours consider this submission not significant.

Major concerns

1.     Various range of concentration of fulleronols should be tested for survivals, locomotion on male and female.

2.     Brain staining with vacuoles should be included.

3.     The graph show that neurons are reduced by the chemicals. But the brain staining is not convincing.

Minor concerns

At 98,  (OOO Monitoring). --- ooo ?

at 122,   0,2 mg/ml ---0.2

A, B at figure 2 should be aligned well with figures

at 226 ,   ### p --- ***p

greater toxicity that С60(OH)30 --- than

at 183, 1 – 0,1; 2 – 0,01; 3 – 0,001; 4 – 0,0001. C - in absence (7) and presence of C60(OH)30, 183 mg/ml: 1 – 2.0; 2 – 0,2; 3 – 0,1; 4 – 0,01; 5 – 0,001; 6 – 0,0001 --- , should be .

at 187, --- autoxidation ?

at 189,  0,2 mg/ml a --- 0.2

at 200 и НОО˙ groups on the carbon cage. --- и ?

at 233, Thus, fullerenols were shown one time more to be safe for flies. --- what does this sentence mean ?

at 238, Fullerenols are active scavengers of ROS, unlike some other 238 antioxidants that imitation the activity of antioxidant defense enzymes. --- English ?

at 303, the third age larvae --- third age ? third instar ?

at 304, They demonstrated of both 304 Cells 2022, 11, x FOR PEER REVIEW 11 of 15 HTT.16Q and HTT.128Q expression resulted in a full-length ~350 kDa human protein in 305 the flies heads. --- English correction

at 311 on Ca2+ -dependent--- Ca2+

Author Response

(The authors gave the same response as above.)

Round 2

Reviewer 1 Report

The author addressed all my comments and thanks for providing additional information.

Reviewer 2 Report

cells (ISSN 2073-4409)

Review of an article: 

«Fullerenols prevent neuron death and reduced oxidative stress in Drosophila Huntington’s disease model»

by Olga I. Bolshakova, Alina A. Borisenkova, Ilya M. Golomidov, Artem E. Komissarov, Alexandra D. Slobodina, Elena V. Ryabova, Irina S. Ryabokon, Evgenia M. Latypova, Elizaveta E. Slepneva, Svetlana V Sarantseva

In Cells (ISSN 2073-4409).

Round 2

The authors addressed well my comments. I would recommend its acceptance.

Reviewer 3 Report

It is argued that C60 ameliorated neurodegeneration, but the brain paraffin stainings are not convincing. Even if considering it convincing, it is not consistent with toxic lifespan  with the C60 feeding.